# NEMOTRON-RESEARCH-TOOL-N1: EXPLORING TOOL-USING LANGUAGE MODELS WITH REINFORCED REASONING

**Shaokun Zhang**[1,2]   **Yi Dong**[1]   **Jieyu Zhang**[3]   **Jan Kautz**[1]   **Bryan Catanzaro**[1]   **Andrew Tao**[1]
**Qingyun Wu**[2]   **Zhiding Yu**[1]   **Guilin Liu**[1]

[1]NVIDIA
[2]Pennsylvania State University
[3]University of Washington

## ABSTRACT

Enabling large language models with external tools has become a pivotal strategy for extending their functionality beyond text space. To enhance LLMs' tool-calling abilities, previous approaches primarily rely on supervised fine-tuning (SFT) with trajectories distilled from stronger models, often resulting in imitative reasoning that limits generalization. In this work, we explore rule-based reinforcement learning (Guo et al., 2025) to enhance tool-calling in LLMs, resulting in Nemotron-Research-Tool-N1, a series of tool-calling reasoning models. Rather than enforcing supervision over intermediate distilled reasoning traces, Tool-N1[1] is trained with a binary RL reward that assesses only the format validity and functional correctness of tool invocations. This lightweight supervision allows the model to develop reasoning strategies independently, without relying on annotated trajectories. Experiments on several major benchmarks show that Tool-N1-7B/14B clearly outperform GPT-4o. We conduct a systematic study on the design of rule-based reinforcement learning strategies for training tool-calling models. Using 5,518 distilled reasoning trajectories, we compare SFT, RL, and the SFT-then-RL pipeline, finding that the widely adopted SFT-then-RL paradigm does not necessarily outperform pure RL. We have open-sourced our implementation here.

## 1 INTRODUCTION

Recent years, equipping large language models with external tools or functions has attracted significant attention, demonstrating impressive performance across a wide range of domains (Bran et al., 2023; Qu et al., 2025). For instance, LLMs augmented with search engines can answer questions that go beyond their training data (Komeili et al., 2021), while those equipped with Python interpreters are capable of solving complex mathematical problems by leveraging external libraries (Wu et al., 2023). These tools effectively enable LLMs to operate beyond purely textual tasks, substantially extending their functional capabilities.

To enhance LLMs tool-calling capability, existing research has primarily focused on synthesizing large volumes of tool-use trajectories using advanced LLMs (Zhang et al., 2024b; Liu et al., 2024a), followed by supervised fine-tuning (SFT) on the generated data. Although step-by-step reasoning has been shown to play a critical role in enabling LLMs to solve complex tasks (Wei et al., 2022), these synthetic datasets often lack explicit reasoning steps. Consequently, current pipelines typically supervise only the tool calls step, without providing guidance on the underlying reasoning process during training. In many cases, reasoning is either omitted entirely in training stage or deferred to the inference stage via prompting techniques (Yao et al., 2023b). While one workaround is to distill reasoning trajectories from advanced models and then train student models via SFT (Chen et al., 2025b), this approach often yields pseudo reasoning: models merely learn to mimic surface-level patterns without truly internalizing the decision-making process (Chen et al., 2025b).

---

[1]Throughout this paper, we refer to Nemotron-Research-Tool-N1 as Tool-N1 for brevity.

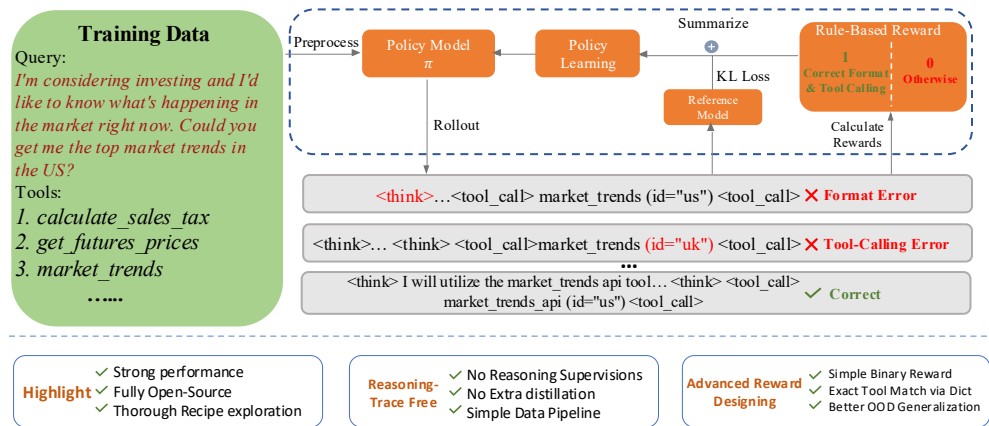

Figure 1: **Overview of the training pipeline for Nemotron-Research-Tool-N1 (Tool-N1).** Starting from standard SFT tool-calling data comprising user queries and candidate tools, we train LLMs to produce structured reasoning and tool calls using a binary reward function within the GRPO algorithm. As supervision is only applied to the format and tool-call correctness, the training process does not require curated reasoning trajectories.

Recently, simple rule-based R1-style reinforcement learning (Guo et al., 2025) has been shown to substantially enhance the complex reasoning capabilities of LLMs (Shen et al., 2025; Ma et al., 2025; Lu et al., 2025b). In these R1-style RL training, rewards are assigned based on the correctness of the final answer and output format, allowing the model to learn the intermediate reasoning steps without explicit reasoning supervision. This paradigm naturally inspired us to explore the utilization of these rule-based RL in training tool call models, for the following reasons: **(1)** Rule-based reward design provides interpretable supervision signals by verifying the correctness of tool calls. In contrast to SFT, which relies on exact next-token prediction and enforces strict output matching, RL allows for more flexibility. Semantically equivalent tool calls, such as those with reordered arguments, can still be rewarded correctly. This enables models to generalize beyond rigid string-level imitation. **(2)** R1-style RL does not require tool-call data with explicit reasoning annotations. Instead, it imposes minimal constraints, requiring only that the model follow a structured output format. This makes it possible to train reasoning capabilities directly from available SFT data. As a result, the model can acquire reasoning skills without curating reasoning traces.

Building on these insights, we address two key questions in this work: *Can rule-based RL be effectively applied to train tool-calling models? How should the RL pipeline be designed?* To this end, we conduct a systematic study on the design of rule-based reinforcement learning strategies for tool-call training, resulting in Nemotron-Research-Tool-N1 **(Tool-N1)**, a series of reasoning tool-using LLMs trained with rule-based RL. The training enforces a simple-tructured reasoning-action format, guiding the model to produce explicit reasoning before invoking tools. We employ a binary reward function that evaluates the correctness of both the reasoning format and tool invocations. This reward design provides precise yet flexible supervision, allowing variation in argument ordering while ensuring functional correctness. The training procedure is shown in Figure 1.

We conduct extensive experiments on BFCL (Yan et al., 2024), API-Bank (Li et al., 2023) and ACEBench (Chen et al., 2025a). Empirical results demonstrate Tool-N1-7B/14B, built upon Qwen2.5-7B/14B-Instruct, clearly outperform GPT-4o and other strong baselines. For instance, Tool-N1-14B achieves approximately 2% and 5% average performance improvements on BFCL and APIBank than GPT-4o. The smaller model, Tool-N1-7B, also surpasses the closed-source GPT-4o and the specialized fine-tuning model Hammer2.1-7B on BFCL by 0.85% and 2.97%, respectively. We further conduct extensive experiments to analyze various aspects of the proposed methodology, including reward design and training data composition. In addition, we curate a dataset of 5,518 tool-calling reasoning trajectories and perform a systematic study on the roles of SFT, RL, and their combinations. Notably, we find that the widely adopted "SFT-then-RL" paradigm does not necessarily yield better performance than pure RL. To summarize, this work makes the following key contributions: **(1)** We propose a simple yet powerful rule-based RL algorithm that converts standard tool-calling data into fully verifiable training signals. This provides a new training paradigm that enables reinforced

reasoning without requiring curated reasoning trajectories. **(2)** Building upon this paradigm, we train Tool-N1-7B/14B and demonstrate strong empirical performance across BFCL, API-Bank, and ACEBench, outperforming GPT-4o and leading open-source baselines. **(3)** We conduct extensive experiments to disentangle the roles of supervised fine-tuning, pure RL, and the widely adopted SFT-then-RL paradigam. Our findings reveal that the commonly used SFT-then-RL paradigm does not always outperform pure RL in tool-calling scenario.

## 2  PROBLEM FORMULATION

We first provide the problem formulation. Consider a LLM and a set of external tools $\mathcal{Z} = \{z_i\}_{i=i}^{I}$ that the LLM can access. Each tool $z_i$ could be represented as a 3-tuple $(n_i, d_i, k_i)$ which includes essential information for tool usage: $n_i$ denotes the tool's name, $d_i$ provides a natural language descriptions, and $k_i$ specifies the tool's input parameters instructions. The model's objective is to respond to user queries in accordance with a policy $\pi$. To achieve this, the LLM may issue multiple tool calls throughout the interaction, each with appropriate input parameters.

At any decision step $t$, the LLM receives two types of input: **(1)** the historical context $c_t$, which consists of all preceding tool-call and observation pairs, and **(2)** the set of currently available tools $\mathcal{Z}$ that can be utilized at this step. The LLM must then decide on the next action. Formally, the decision-making process is defined as: $\pi(c_t, \tilde{\mathcal{Z}}) \to a_t,$  s.t.  $a_t \subseteq \mathcal{Z}$, where $a_t$ denotes the action selected at step $t$. This action corresponds to one or more tool calls drawn from the accessible tool subset $\tilde{\mathcal{Z}}$. $c_t$ represents the historical context. Specifically,

$$\begin{cases} a_t = \{z_0(p_0), \ldots, z_m(p_m)\}, \\ c_t = (a_0, o_0, \ldots, a_t, o_t) \end{cases} \tag{1}$$

where each $z_m$ denotes the $m$-th tool invoked and $p_m$ its corresponding parameters. The value of $m$ indicates the number of tool calls made at time $t$. We employ $o_t$ denotes the observation after action $a_t$ is taken. The ultimate goal of tool learning is to enable LLMs with a generalized policy $\pi$ that effectively addresses user queries by producing a sequence of action-observation pairs $(a_t, o_t)$.

## 3  NEMOTRON-RESEARCH-TOOL-N1

In this subsection, we explore the use of R1-style reinforcement learning for training tool-calling language models, resulting in Nemotron-Research-Tool-N1, a series of general-purpose tool-using reasoning LLMs. Tool-N1 is built on the GRPO algorithm (Guo et al., 2025; Shao et al., 2024), aiming to enhance tool-calling capabilities with reasoning in complex scenarios where the LLM must solve queries using a set of accessible tools. We further conduct a systematic study on the design of rule-based reinforcement learning strategies for this training paradigm.

Formally, given the historical context $c_t$ and the set of currently available tools $\mathcal{Z}$, the model generates a set of candidate responses $[O^1, O^2, ..., O^N]$, where each $O^n \in \mathcal{O}$. Here $\mathcal{O}$ denotes the space of possible output responses, each comprising: **(1)** textual reasoning. **(2)** an associated action $a_n$. These responses are evaluated using a reward function, yielding a reward set $\{r_1, r_2, ..., r_N\}$. We then optimize the policy $\pi_\theta$ using GRPO, the optimization objective is shown below:

$$\mathcal{L}_{\text{GRPO}}(\theta) = \mathbb{E}_{(c_t, \mathcal{Z})} \mathbb{E}_{O^i \sim \mathcal{O}} \Big[ \min \Big( \rho_i A_i, \ \text{clip}(\rho_i, 1 - \epsilon, 1 + \epsilon) A_i \Big)$$
$$- \beta \, \text{KL}\big(\pi_\theta \, \| \, \pi_{\text{old}}\big) \Big], \text{where } \rho_i = \frac{\pi_\theta(O^i \mid c_t, \mathcal{Z})}{\pi_{\text{old}}(O^i \mid c_t, \mathcal{Z})}. \tag{2}$$

$\epsilon$ and $\beta$ represent tunable hyperparameters. $A_i$ represents the relative advantage of the $i$-th response, which is computed as follows:

$$A_i = \frac{r_i - \text{mean}(\{r_1, r_2, \ldots, r_N\})}{\text{std}(\{r_1, r_2, \ldots, r_N\})}, \tag{3}$$

where $mean$ and $std$ represent the mean and standard deviation of the rewards, respectively. In subsequent sections, we detail the following components of our approach: (1) the preparation of tool-calling data and its integration into the RL training pipeline, (2) the structured reasoning template adopted during training, and (3) the reward modeling strategy used to compute the reward signal $r$.

## 3.1 DATA PREPARATION

Numerous prior works have focused on collecting large-scale tool-calling trajectories (Liu et al., 2024a; Zhang et al., 2024b; Yin et al., 2025; Qin et al., 2023), followed by supervised fine-tuning to improve LLMs' tool-use capabilities. Each instance in these dataset typically consist of natural language user query $Q$, paired with a sequence of ground-truth tool invocation steps in the form $(a_0, o_o, \ldots, a_t, o_t)$. The model is then trained to predict each subsequent action $a_{t+1}$, conditioned on the observed trajectory up to that point. However, SFT often exhibits limited generalization ability, as the model tends to memorize the training trajectories rather than developing robust, intrinsic reasoning capabilities which has been pointed out in recent work (Chen et al., 2025b). To fully leverage the available tool-calling data in the community, we unify and preprocess data from xLAM (Zhang et al., 2024b) and a subset of ToolACE (Liu et al., 2024a), which provide single-turn and multi-turn synthetic tool-calling trajectories. As these datasets are generated by potentially unstable LLMs, they often contain inconsistencies and unstructured formats unsuitable for RL training.

We standardize the data by filtering out samples with invalid tool calls, specifically, those involving tools absent from the candidate tool list. Available tools are extracted from the system prompt, and both candidate and ground-truth tools are parsed into structured dictionary formats. Instances that fail JSON parsing or contain formatting inconsistencies are discarded. We ensure this processing yields a clean and consistent data suitable for reinforcement learning. For multi-turn data from the ToolACE, we further segment each trajectory into multiple single-step prediction instances, where each instance contains one target tool call and the preceding steps are treated as context. We train LLMs using R1-style GRPO to predict each tool invocation step based on this contextual information and provided tools. We present the statistic information of the proceed data in Table 5 in Appendix C.

## 3.2 THINKING TEMPLATE



**Thinking Template**

Here is a list of functions in **JSON** format that you can invoke: <tools> {tools} </tools>.
**In each action step, you MUST:**
**1.** Think about the reasoning process in the mind and enclosed your reasoning within <think></think> XML tags.
**2.** Then, provide a json object with function names and arguments within <tool_call> </tool_call> XML tags. i.e., <tool_call>["name": <function-name>, "arguments": <args-json-object>, "name": <function -name2>, "arguments": <args-json-object2>, ...]</tool_call>
**3.** Make sure both the reasoning and the tool call steps are included together in one single reply.
**A complete reply example is:** <think>To address the query, I need to send the email to Bob and then buy the banana through walmart.</think> <tool_call> ["name":"email", "arguments":"receiver": "Bob", "content": "I will bug banana through walmart", "name": "walmart", "arguments": "input": "banana"]</tool_call>. Please make sure the type of the arguments is correct.



Figure 2: The resoning prompt template used during training and inference. The prompt guides the LLM to explicitly separate its internal reasoning (within <think> tags) from tool invocation actions (within <tool_call> tags). We show the full prompt template in Appendix F.

Following prior work (Guo et al., 2025), we adopt a lightweight prompting template to elicit tool calls from the LLM which is shown in Figure 2. The prompt explicitly instructs the model to generate intermediate reasoning within <think>...</think> tags, followed by the tool invocation enclosed in <tool_call>...</tool_call> tags. The design philosophy behind this template is to minimize reliance on overly rigid formatting rules, which could reducing the risk of overfitting to specific prompt patterns. By allowing greater flexibility in how the model expresses its reasoning, we aim to promote more robust generalization across diverse tool-use scenarios. Additionally, the use of this lightweight prompting design during training enables the resulting model to be more easily integrated with more sophisticated prompting strategies (Yao et al., 2023b;a; Zhang et al., 2024c).

## 3.3 REWARD MODELING

Following the data preparation described in Section 3.1, we construct a training dataset in which each ground-truth tool call is represented as a structured dictionary. This format enables reliable verification of tool names and argument-value pairs during reinforcement learning rather than simply string match. Leveraging this structure, we define a binary reward function in which jointly evaluates the correctness of the reasoning format and the accuracy of the tool call.

**Format Checking.** Following prior work (Guo et al., 2025; Shao et al., 2024; Meng et al., 2025), we incorporate format checking during training to verify whether the model's output adheres to the expected structural conventions, specifically, whether the reasoning is enclosed within `<think>...</think>` tags and the tool call is properly placed within `<tool_call>...</tool_call>` tags. This structural constraint encourages the model to engage in explicit reasoning before tool calling, rather than shortcutting to the final answer. By enforcing format adherence, we aim to cultivate the model's intrinsic reasoning ability, which may potentially contributes to improved generalization, particularly on out-of-distribution inputs.

**Tool-Calling Checking.** We also check the correctness of the tool call. The tool-call output is parsed as dictionaries, enabling exact matching against the ground-truth. This involves checking whether the predicted tool name matches the ground-truth and whether all required arguments are present correctly. This strict matching criterion ensures that the model learns to generate functionally precise and executable tool calls. Compared to the next-token prediction logic in SFT, this dictionary-based matching introduces greater flexibility. It allows argument order to vary without penalization, encouraging the model to focus on the underlying semantics rather than surface-level memorization.

**Binary Reward Definition.** Given previous context $c_t$ and the LLM output $O_t$, we define a binary reward function $r(c_t, O_t) \in \{0, 1\}$ that assigns a reward of 1 if both of the following conditions are met: **(1) Format Correctness:** The model output adheres to the structural format, i.e., contains both `<think>...</think>` and `<tool_call>...</tool_call>` tags. **(2) Tool Call Correctness:** The predicted tool call $a_t \in O_t$ exactly matches the ground-truth call $a_t^*$ in both tool name and all argument key-value pairs.

$$r(c_t, O_t) = \begin{cases} 1, & \text{if } \texttt{FormatCorrect}(O_t) \land \texttt{ToolCallMatch}(a_t, a_t^*) \\ 0, & \text{otherwise} \end{cases} \tag{4}$$

where $\texttt{FormatCorrect}(O_t)$ returns true if the output is correctly wrapped in both required tags, and the $\texttt{ToolCallMatch}(a_t, a_t^*)$ returns true if $a_t$ matches the ground-truth tool call $a_t^*$ exactly in structure and content. We find that a simple binary reward scheme is most effective for training tool-calling LLMs, as validated through extensive ablation studies presented in Section 4.4.1.

## 4 EXPERIMENTS

We conduct experiments to prove the superiority of the proposed method. We begin by providing the experimental settings in Section 4.1. We then evaluate the training method representative benchmarks to verify its effectiveness. Finally, we perform in-depth investigations for the training paradigam.

### 4.1 SETTINGS

**Datasets.** We primarily utilize a subsets of ToolACE (Liu et al., 2024a) and xLAM (Zhang et al., 2024b) as our training data. ToolACE (Liu et al., 2024a) encompasses a wide range of tool-calling scenarios, including examples with multiple candidate tools and parallel tool calls, and covers a pool of 26,507 diverse tools. In contrast, xLAM focuses on single-turn tool calling, comprising 60,000 instances collected through APIGen (Liu et al., 2024b).

**Models.** Unless otherwise noted, we use Qwen2.5-7B/14B-Instruct (Yang et al., 2024) as the primary backbone model throughout this work. To assess the generalization ability of our method, we also perform evaluations on alternative backbone models, including multiple variants from the LLaMA family in different scales. In our experiments, we compare against both general-purpose open-source models, such as the GPT series and Gemini-2.0, as well as specialized tool-calling models, including ToolACE-8B (Liu et al., 2024a), xLAM-2 (Prabhakar et al., 2025), and Hammer2.1 (Lin et al., 2024).

**Benchmarks.** We primarily evaluate the performance on single-turn tool-calling scenarios with several representative benchmarks, including the Berkeley Function Call Leaderboard (BFCL) (Yan et al., 2024), API-Bank (Li et al., 2023) and ACEBench (Chen et al., 2025a). For BFCL, we conduct evaluations on both the Non-live and Live subsets, corresponding to synthetic and real-world data, respectively. Each subset includes four categories: Simple, Multiple, Parallel, and Parallel Multiple. Simple and Multiple scenarios both involve the invocation of a single tool, with the Multiple category featuring multiple candidate tools. In contrast, Parallel and Parallel Multiple scenarios require the simultaneous invocation of multiple tools. For API-Bank and ACEBench, we exclude multi-turn cases from our evaluation. Performance across all benchmarks is reported in terms of accuracy.

| Models | Non-Live | | | | Live | | | | Overall | | |
|---|---|---|---|---|---|---|---|---|---|---|---|
| | *Simple* | *Multiple* | *Parallel* | *Parallel Multiple* | *Simple* | *Multiple* | *Parallel* | *Parallel Multiple* | *Non-live* | *Live* | *Overall* |
| **GPT-4o** | 79.42 | 95.50 | 94.00 | 83.50 | **84.88** | 79.77 | 87.50 | 75.00 | 88.10 | 79.83 | 83.97 |
| **GPT-4o-mini** | 80.08 | 90.50 | 89.50 | 87.00 | 81.40 | 76.73 | **93.75** | 79.17 | 86.77 | 76.50 | 81.64 |
| **GPT-3.5-Turbo-0125** | 77.92 | 93.50 | 67.00 | 53.00 | 80.62 | 78.63 | 75.00 | 58.33 | 72.85 | 68.55 | 70.70 |
| **Gemini-2.0-Flash-001** | 74.92 | 89.50 | 86.50 | 87.00 | 75.58 | 73.12 | 81.25 | **83.33** | 84.48 | 81.39 | 82.94 |
| **DeepSeek-R1** | 76.42 | 94.50 | 90.05 | 88.00 | 84.11 | 79.87 | 87.50 | 70.83 | 87.35 | 74.41 | 80.88 |
| **Llama3.1-70B-Inst** | 77.92 | **96.00** | 94.50 | 91.50 | 78.29 | 76.16 | 87.50 | 66.67 | 89.98 | 62.24 | 76.11 |
| **Llama3.1-8B-Inst** | 72.83 | 93.50 | 87.00 | 83.50 | 74.03 | 73.31 | 56.25 | 54.17 | 84.21 | 61.08 | 72.65 |
| **Qwen2.5-7B-Inst** | 75.33 | 94.50 | 91.50 | 84.50 | 76.74 | 74.93 | 62.50 | 70.83 | 86.46 | 67.44 | 76.95 |
| **xLAM-2-70b-fc-r (FC)** | 78.25 | 94.50 | 92.00 | 89.00 | 77.13 | 71.13 | 68.50 | 58.33 | 88.44 | 72.95 | 80.70 |
| **ToolACE-8B (FC)** | 76.67 | 93.50 | 90.50 | 89.50 | 73.26 | 76.73 | 81.25 | 70.83 | 87.54 | 78.59 | 82.57 |
| **Hammer2.1-7B (FC)** | 78.08 | 95.00 | 93.50 | 88.00 | 76.74 | 77.4 | 81.25 | 70.83 | 88.65 | 75.11 | 81.88 |
| **Tool-N1-7B** | 77.00 | 95.00 | 94.50 | 90.50 | 82.17 | 80.44 | 62.50 | 70.83 | 89.25 | 80.38 | 84.82 |
| **Tool-N1-14B** | **80.58** | **96.00** | 93.50 | **92.00** | 84.10 | **81.10** | 81.25 | 66.67 | **90.52** | **81.42** | **85.97** |

Table 1: Comparison on the BFCL benchmark (last updated on 2025-04-13). Average performance is calculated using the official script. The best results in each category are highlighted in **bold**, while the second-best results are underlined.

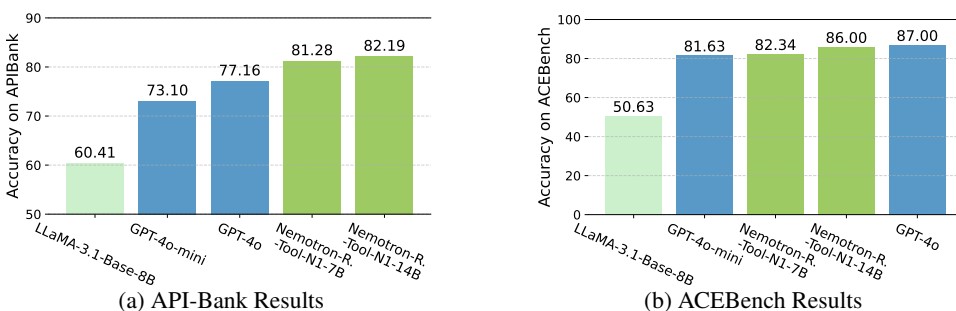

(a) API-Bank Results          (b) ACEBench Results

Figure 3: Performance comparison on two additional benchmarks, ACEBench and API-Bank.

**Other Implementation Details.** We conduct all RL training using the open-source reinforcement learning library Verl (Sheng et al., 2024). Training is performed with a batch size of 1024 and a learning rate of $1 \times 10^{-6}$. The temperature is fixed at 0.7. We set the entropy coefficient to 0, as we observe that introducing entropy negatively impacts exploration. The KL divergence loss coefficient is set to $1 \times 10^{-3}$ across all experiments. All training runs are executed on a cluster of 4 nodes, each equipped with 8 NVIDIA H100 80GB GPUs. For SFT training, we employ LLaMA-Factory (Zheng et al., 2024a). More details of the implementations could be found in Appendix E.

## 4.2 MAIN RESULTS

**Results on BFCL.** Table 1 presents the evaluation results on the BFCL benchmark, including detailed accuracy for each subcategory. We adopt the official evaluation script from the BFCL leaderboard and report the average accuracy across categories. We observe that specialized fine-tuned models, such as ToolACE-8B and Hammer-2.1-7B, benefit significantly from domain-specific training data, achieving better performance than GPT-4o-mini despite their smaller model sizes. However, neither surpasses GPT-4o. In contrast, Tool-N1-7B and Tool-N1-14B models achieve clearly superior overall performance, outperforming both state-of-the-art closed-source models (e.g., GPT-4o) and domain-specialized fine-tuned models (e.g., xLAM-2-70B and ToolACE-8B). Notably, our trained tool-call reasoning models significantly outperform supervised fine-tuning (SFT) baselines trained on the same data sources, including the ToolACE (Liu et al., 2024a) and xLAM (Zhang et al., 2024b) series. These results collectively demonstrate that the R1-style reinforcement learning paradigm provides a more effective approach for improving the tool-calling capabilities of large language models.

**Results on More Benchmarks.** To provide a more comprehensive evaluation, we conduct experiments on two additional benchmarks: ACEBench (Chen et al., 2025a) and APIBank (Li et al., 2023). The results are shown in Figure 3. On these benchmarks, Tool-N1-7B achieves substantial absolute improvements of over 20% on APIBank and over 30% on ACEBench compared to the base models. Notably, Tool-N1-7B outperforms GPT-4o-mini on both benchmarks, while Tool-N1-14B surpasses GPT-4o by 5.03%. These results demonstrate that models trained using our proposed

method generalize effectively across diverse tool-calling scenarios, thereby opening new opportunities for scaling performance in LLM tool-augmented learning.

## 4.3 DEEP ANALYSIS

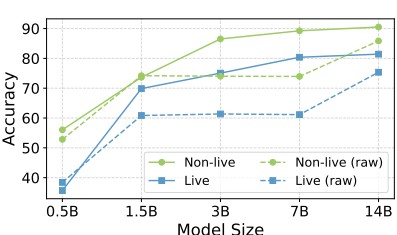 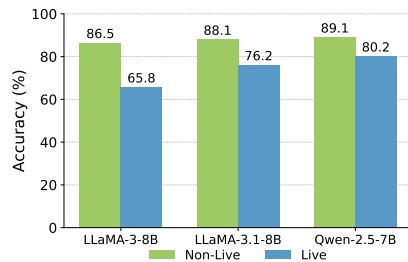

Figure 4: Scaling performance across model sizes using the Qwen2.5-Instruct series. We report results for both RL-trained and original instruction-tuned models without further post-training.

Figure 5: Performance across different backbones. All models use their instruction-tuned variants. At comparable scales, Qwen consistently outperforms both LLaMA 3 and LLaMA 3.1.

### 4.3.1 SCALABILITY AND GENERALIZABILITY

**Scalability.** The scaling law, which characterizes the relationship between model size and performance, plays a critical role in understanding the effectiveness of the training methods. We assess the scaling behavior of the proposed training method by evaluating a range of model sizes, including 0.5B, 1.5B, 3B, 7B and 14B from the Qwen2.5-Instruct series. For comparison, we also report the performance of the original instruction-tuned models without any additional training. We report average performance on both the Live and Non-Live categories of the BFCL benchmark, with detailed results presented in Figure 4. As expected, larger models consistently outperform smaller ones in both evaluation settings. Notably, performance improvements from post-training are limited for smaller models (0.5B and 1.5B), whereas larger models exhibit substantial gains. These findings suggest that the R1-style training method scales more effectively with increasing model size.

**Generalizability.** We further evaluate the impact of LLM backbones Touvron et al. (2023) to investigate the generalization capability of the proposed method. In addition to the Qwen, we include experiments using LLaMA series: LLaMA3-8B-Instruct and LLaMA3.1-8B-Instruct. These evaluations are conducted on the BFCL benchmark, with results presented in Figure 5. Our findings show that Qwen2.5-Instruct significantly outperforms both LLaMA variants at the same model scale. This advantage is likely due to Qwen's inherently stronger reasoning capabilities, as observed by Gandhi et al. (2025). As a result, the R1-style training paradigm is able to elicit better performance when applied to Qwen series LLMs.

> **Finding 1**
>
> The R1-style training method scales effectively with model size, yielding significantly greater performance gains for larger models compared to smaller ones. It also generalizes well across backbones, with Qwen models outperforming LLaMA variants at the same scale.

### 4.3.2 SFT OR RL? A DEEP INVESTIGATION INTO THE TRAINING RECIPES

| Method | No-Reason SFT | | Reason-SFT | | Reason-SFT+RL | RL | |
|---|---|---|---|---|---|---|---|
| Recipes | 50% SFT | 100% SFT | 50% SFT | 100% SFT | 50% SFT+50%RL | 50% RL | 100%RL |
| Non-Live | 85.94 | 86.40 | 86.40 | 87.54 | 88.19 | 87.12 | 88.23 |
| Live | 76.09 | 76.54 | 76.61 | 77.87 | 78.16 | 78.90 | 78.24 |
| Avg | 81.02 | 81.47 | 81.50 | 82.71 | 83.17 | 83.01 | **83.24** |

Table 2: Performance comparison of different training recipes on tool-calling tasks with 5,518 instances from ToolACE. "No-Reason SFT" fine-tunes on tool-call data without reasoning traces; "Reason-SFT" uses reasoning trajectories; "Reason-SFT+RL" applies RL after Reason-SFT; Percentages indicate the proportion of data used for each stage.

Although the proposed R1-style reinforcement learning paradigm demonstrates strong performance, we conduct a more comprehensive comparison against SFT, including both SFT with and without reasoning trajectories. Our goal is to identify the most effective training recipe for tool-using LLMs

with reasoning capabilities. To ensure fair comparison, we filter out single-turn data from ToolACE and distilled reasoning trajectories using DeepSeek-R1 (Guo et al., 2025), one of the most advanced tool-using language models. We prompt DeepSeek-R1 to generate responses with reasoning enclosed in <think>...</think> tags and tool calls in <tool_call>...</tool_call> tags. Only trajectories that (1) follow the correct format and (2) produce correct tool calls are retained, yielding 5,518 high-quality instances. All the experiments are performed upon Qwen2.5-Instruct model.

Using these data, we construct several training recipes: **(1) Reason-SFT**, which performs SFT on distilled reasoning trajectories; **(2) No-Reason SFT**, which performs SFT only on corresponding data without reasoning traces; **(3) RL**, the proposed RL training method. **(4) Reason-SFT+RL**, which applies SFT followed by RL on the same dataset. We could observe that: **(1)** Although the combination of SFT on reasoning trajectories followed by RL is commonly regarded as the best practice in many domains, we do not observe improved performance under equal data budgets in the tool-calling setting (83.17% vs. 83.24%). In fact, SFT may even hinder performance, a phenomenon also noted by Chen et al. (2025b). **(2)** Pure RL outperforms both Reason-SFT and No-Reason SFT under equal data budgets, achieving the best performance (83.24% with 100% RL vs. 82.71% with 100% Reason-SFT and 81.47% with 100% No-Reason SFT). This finding is consistent with prior observations Chen et al. (2025b), which suggest that SFT may encourage pseudo-reasoning rather than genuine task-oriented reasoning. **(3)** Interestingly, No-Reason SFT performs only slightly worse than Reason-SFT (e.g., 81.47% vs. 82.71%), suggesting that providing reasoning traces during SFT offers limited additional benefit.

> **Finding 2**
>
> The commonly adopted SFT-then-RL pipeline does not necessarily outperform pure RL in tool-calling tasks. Notably, pure RL, trained without reasoning distillation data, demonstrates stronger performance than any other combinations involving SFT and RL.

## 4.4 VISUALIZATION

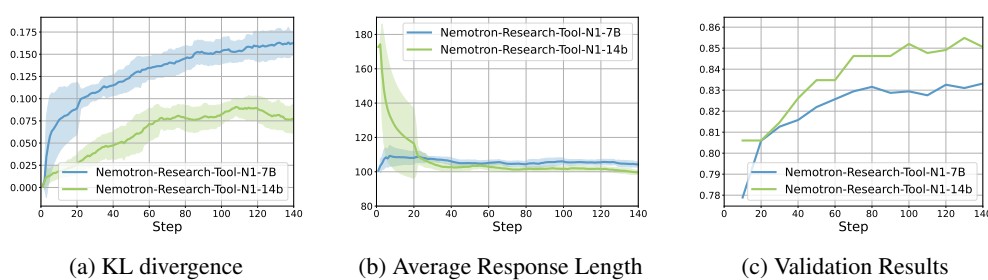

(a) KL divergence     (b) Average Response Length     (c) Validation Results

Figure 6: Learning curves across training steps for Tool-N1-7B and Tool-N1-14B. We report KL divergence, average response length, and validation accuracy.

We present additional visualizations of the training dynamics in Figure 6, including KL divergence, average response length, and validation accuracy over training steps. The KL divergence and validation accuracy, shown in Figures 6a and 6c, exhibit consistent improvement, suggesting that the model get stable learning and exploration under the training setup. Notably, Figure 6b shows no significant increase in response length throughout training, which contrasts with trends observed in prior R1-style training works (Guo et al., 2025; Liu & Zhang, 2025; Shen et al., 2025). Tool-N1-7B maintains a relatively flat trajectory from the beginning, while Tool-N1-14B gradually decreases to a comparable length. This observation suggests that longer reasoning chains do not necessarily lead to better tool-use performance; instead, there appears to be an vertain response length that could be sufficient for accurate tool calling.

### 4.4.1 ABLATIONS

**Ablations on Reward Design.** To assess how reward granularity affects model behavior, we evaluate Tool-N1-7B under two reward schemes: fine-grained and binary (Table 3). The fine-grained setting provides partial reward, 0.2 for correct reasoning format and an additional 0.2 for matching function names, even if the final function call is incorrect. In contrast, the binary setting only gives a reward of 1.0 when all components are correct, including reasoning, function name, and arguments. Here, w/ Reason Format requires the model to output intermediate reasoning in a specific format, while w/o removes this constraint. The results are shown in Table 3. Tool-N1 achieves consistently better

| | Fine-Grained Reward | | Binary Reward | |
|---|---|---|---|---|
| Split | w/ Reason Format Partial Reward | w/ Reason Format + Func name Partial Rewards | w/o Reason Format | w/ Reason Format |
| Non-Live | 87.83 | 88.54 | 87.63 | 89.25 |
| Live | 79.64 | 76.61 | 76.24 | 80.38 |
| Avg | 83.74 | 82.58 | 81.94 | 84.82 |

Table 3: Ablation study on reward granularity. We compare fine-grained reward designs, where partial credit is given for correct reasoning format and correct function names in function call stages, with binary rewards that assign full reward only when all conditions are fully satisfied. The results show that binary rewards consistently yield better performance, especially in the Live setting.

performance with binary rewards, particularly on the Live subset (80.38% vs. 76.61%), which involves more realistic inputs. We attribute this to reduced reward hacking (Pan et al., 2024): under fine-grained schemes, the model may overfit to superficial cues such as formatting or partial matches, without ensuring full execution correctness. Furthermore, within the binary setup, we observe that removing the reasoning format constraint significantly hurts performance (dropping from 80.38% to 76.24%). This highlights the critical role of structured reasoning in guiding Tool-N1-7B toward reliable and generalizable tool use, especially in complex, real-world scenarios.

> **Finding 3**
>
> Binary rewards lead to superior performance compared to fine-grained rewards, especially on realistic inputs. Additionally, enforcing structured reasoning is crucial, removing it significantly degrades performance, underscoring its importance in guiding reliable tool use.

**Ablations on Training Data Composition.**

| Recipe | Raw Model | xLAM 8B | ToolACE 8B | xLAM w/o ToolACE | ToolACE w/o xLAM | ToolACE w/ xLAM |
|---|---|---|---|---|---|---|
| Non-Live | 73.94 | 84.40 | 87.54 | 87.77 | 87.67 | 89.25 |
| Live | 61.14 | 66.90 | 78.59 | 76.24 | 79.58 | 80.38 |
| Avg | 67.54 | 75.65 | 82.57 | 82.01 | 83.63 | 84.82 |

Table 4: Effect of training data composition on Tool-N1-7B performance. We compare the full training recipe against ablated variants that exclude xLAM or ToolACE data. Results show that both sources contribute to performance, with ToolACE providing greater gains.

We then investigate how different data composition strategies affect performance on the BFCL benchmark. Experiments are conducted using the Tool-N1-7B model, with results presented in Table 4. Our key findings are as follows: (1) ToolACE data yields particularly strong improvements in the live setting. Overall, these results suggest that progressively enriching the training data enhances the model's tool-calling proficiency. (2) Compared to the raw model (Qwen2.5-7B-Instruct), R1-style training significantly enhances tool-calling capabilities. (3) Compared to models trained using SFT on the same data source, the R1-style training consistently yields better performance. Specifically, the Tool-N1-7B model trained solely on xLAM data outperforms the xLAM-8B SFT model by 6.36%, and the Tool-N1-7B model trained solely on the ToolACE subset exceeds the ToolACE-8B SFT model by 1.62%, despite using only a subset of the data.

## 5 CONCLUSION

We introduce Nemotron-Research-Tool-N1, a series of tool-using language models trained using a rule-based reinforcement learning paradigm. Unlike previous approaches that rely on SFT with annotated reasoning trajectories, Tool-N1 employs a reward function that supervises only the final answer and the structural format of the reasoning. This enables the model to learn effective reasoning strategies without requiring step-by-step annotations. Experimental results demonstrate that Nemotron-Research-Tool-N1 consistently outperforms existing baselines across multiple benchmarks. In addition, we curate a set of 5,518 distilled reasoning trajectories to analyze the effects of supervised fine-tuning, reinforcement learning, and their combination. Our findings indicate that the commonly used SFT-then-RL pipeline does not necessarily yield better performance than pure RL.

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

## A   THE USE OF LARGE LANGUAGE MODELS

We used large language models (LLMs) only for light language polishing (e.g., improving grammar and readability). No LLMs were involved in research ideation, data analysis, or substantive writing.

## B   RELATED WORK

**Tool Learning.** Enhancing large language models (LLMs) with external tools has demonstrated significant potential in addressing complex tasks (Qu et al., 2025; Wang et al., 2024b; Zhang et al., 2024d). Typical applications include integrating LLMs with search engines (Komeili et al., 2021; Lazaridou et al., 2022; Shuster et al., 2022; Zhang et al., 2025), calculator (Nakano et al., 2021), vision tools (Ma et al., 2024b), and Python interpreters (Song et al., 2024; Wang et al., 2024a). Research efforts aimed at improving LLMs' tool-use capabilities can be broadly divided into two categories.

- The first one focuses on improving the backed LLM itself. This typically involves curating large-scale supervised datasets (Qin et al., 2023; Zhang et al., 2024b; Liu et al., 2024a) and applying either SFT (Qin et al., 2023; Zhang et al., 2024b; Liu et al., 2024a; Zeng et al., 2025a; Abdelaziz et al., 2024; Chen et al., 2024a;b) or DPO reinforcement learning (Zeng et al., 2025b). Although these approaches may bring performance gains, they are fundamentally data-centric (Zha et al., 2025) and rely heavily on access to high-quality, large-scale training corpora. Furthermore, enabling reasoning capabilities requires distilling reasoning trajectories from advanced models, which introduces additional costs.

- The second category explores prompting-based techniques that do not alter the underlying model. These methods aim to elicit tool-use behavior through strategies such as in-context learning with demonstrations (Paranjape et al., 2023; Kim et al., 2023; Liu et al., 2024b; Zhang et al., 2024a) or the design of advanced prompting and workflow schemes (Yao et al., 2023b; Zhang et al., 2024c; Lu et al., 2025a; Son et al., 2025; Schick et al., 2023; Fawzi et al., 2025; Zheng et al., 2024b; Wu et al., 2025). However, due to the inherent limitations of the base models, such prompting approaches often struggle with tasks that require intricate reasoning or complex multi-step actions. In addition, complex prompting workflows introduce substantial inference overhead.

**LLM Reasoning and Reinforcement Learning.** Recent efforts in the LLM research community have increasingly focused on improving reasoning capabilities, marking a shift from train-time scaling to test-time scaling (Muennighoff et al., 2025), with the goal of enabling models to handle complex problem-solving tasks (Wei et al., 2022) without perfoming heavy-lift modle traingrequiring computationally intensive additional model training Ma et al. (2023; 2024a). Earlier approaches often relied on step-level supervision or learned reward models to guide the model's reasoning trajectory (Gao et al., 2024; Li & Li, 2024). More recently, DeepSeek-R1 (Guo et al., 2025) has demonstrated that simple rule-based reinforcement learning can effectively induce strong reasoning behaviors. In this framework, reward functions are defined based on predefined criteria, i.e., checking whether the model's final answer matches the ground truth in math problems. These rewards target only the correctness of the final answer, allowing the model to learn the intermediate reasoning steps without explicit reasoning supervision. This R1-style reasoning paradigm has shown success across various domains, including mathematics (Shao et al., 2024), coding (Liu & Zhang, 2025), and vision tasks (Shen et al., 2025). In parallel, several recent studies have integrated external tools such as search engines (Jin et al., 2025) and code interpreters (Feng et al., 2025) into reinforcement learning frameworks for LLM reasoning. These approaches primarily focus on training LLMs to effectively utilize a single, general-purpose tool to support one single reasoning chain. In contrast, our work aims to enhance LLMs' tool-calling ability in more general tool-use settings, where multiple tools with complex functionalities and argument structures are available.

## C   DATA DETAILS

In this section, we provide more details of both the training and test dataset.

### C.1   TRAINING DATA

xLAM (Zhang et al., 2024b) is a tool-calling dataset curated for supervised fine-tuning, consisting exclusively of single-turn interactions. Each turn involves either a single-tool call or a multi-tool call.

In contrast, ToolACE includes both single-turn and multi-turn tool-calling interactions, with each turn containing either a single-tool or multi-tool call. After formatting and unifying the data, the final training set statistics are presented in Table 5. More information could be found in Section 3.1.

| | xLAM | | ToolACE | |
|---|---|---|---|---|
| | Single-Turn | Multi-Turn | Single-Turn | Multi-Turn |
| Raw Data | 60000 | 0 | 10500 | 800 |
| After Process | 60000 | 0 | 8183 | 1470 |

Table 5: The statistic information in the training data preparation stage.

## C.2 TEST DATA

We then provide a brief introduction for the benchmark involved in this paper.

**BFCL.** BFCL evaluates the LLM's ability to call functions accurately. The data and leaderboard is consistly evolving and the benchmark is widely-used in the tool learning research. The current version (v3) categorizes the data into three types: Live, Non-Live, and Multi-turn. This paper primarily focuses on the Live and Non-Live categories. Non-Live refers to test cases generated synthetically or curated by researchers, while Live data consists of real user-contributed queries.

**ACEBench.** ACEBench is designed to evaluate tool-use capabilities with fine-grained categorization which could be divided into three primary categories: Normal, Special, and Agent. In this study, we concentrate on two sub-categories within the Normal category: Atom and Single-turn. Atom cases consist of API calls that utilize specific parameter types. The Single-turn subset encompasses both sequential and parallel tool-calling scenarios.

**APIBank.** API-Bank is a tool-calling benchmark with two modes: Call and Retrieve + Call. The model is expected to interpret user intents from the dialogue and execute the corresponding local Python tools. Evaluation is based on whether the tool outputs align with the expected results. In our experiments, we specifically utilize the Call mode.

## D MORE LEARNING CURVES

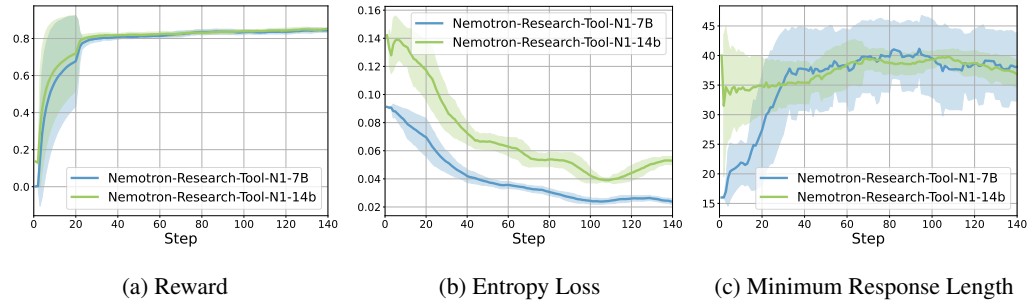

(a) Reward  (b) Entropy Loss  (c) Minimum Response Length

Figure 7: More Learning curves across training steps for Tool-N1-7B and Tool-N1-14B. We report reward, entropy loss, and the minimum response length.

We further analyze the training dynamics in Figure 7. The reward values increase rapidly, while the entropy loss decreases gradually for both models, indicating a stable learning process. Additionally, Figure 7c shows the minimum response lengths over training. We observe that Tool-N1-7B exhibits a steady increase, eventually converging to the value achieved by Tool-N1-14B. In contrast, Tool-N1-14B maintains a relatively stable response length throughout training. This suggests that the smaller model undergoes greater improvements in reasoning ability compared to the larger one.

## E MORE IMPLEMENTATION DETAILS

We conduct RL experiments using the open-source Verl library. We perform hyperparameter tuning via grid search to ensure a stable and efficient training, focusing primarily on the learning rate, KL coefficient, entropy coefficient, number of rollouts, and batch size. The hyperparameters used during

| Hyperparameters | Value | Hyperparameters | Value |
|---|---|---|---|
| Batch_Size | 1024 | Epoch Number | 7 |
| Max Prompt Length | 4096 | KL Coefficient | 1e-3 |
| Max Response Length | 8192 | Entrophy Coefficient | 0 |
| Learning Rate | 1e-6 | Rollout Number | 5 |
| Temperature | 0.7 | Gpu Memory Utilization | 0.6 |

Table 6: The detailed hyperparameters used for RL training.

training are summarized in Table 6. The key hyperparameters used during training are summarized in Table 6. For both Tool-N1-7B and Tool-N1-14B, we adopt the same set of hyperparameters.

## F   PROMPTS

We use identical prompts across model training, evaluation, and reasoning trajectory distillation from the advanced model DeepSeek-R1. The specific prompt is provided below.

---

**Full Prompt Template**

You are an expert in composing functions. You are given a question and a set of possible functions. Based on the question, you will need to make one or more function/tool calls to achieve the purpose. If none of the function can be used, point it out. If the given question lacks the parameters required by the function, also point it out. You should only return the function call in tools call sections.

**# Tool**
Here is a list of functions in **JSON** format that you can invoke:
<tools>
{tools}
</tools>.

**In each action step, you MUST:**
**1.** Think about the reasoning process in the mind and enclosed your reasoning within <think></think> XML tags.
**2.** Then, provide a json object with function names and arguments within <tool_call> </tool_call> XML tags. i.e., <tool_call>["name": <function-name>, "arguments": <args-json-object>, "name": <function -name2>, "arguments": <args-json-object2>, ...]</tool_call>
**3.** Make sure both the reasoning and the tool call steps are included together in one single reply.

A complete reply example is: <think>To address the query, I need to send the email to Bob and then buy the banana through walmart.</think> <tool_call> ["name":"email", "arguments":"receiver": "Bob", "content": "I will bug banana through walmart", "name": "walmart", "arguments": "input": "banana"]</tool_call>. Please make sure the type of the arguments is correct.

---

Figure 8: Full system prompts used during model training and inference.

## G    EXAMPLES

**Example 1: Single Function Calling**

Query: Can you search for a hair stylist in Lafayette in Louisiana for me, please?

Output: <think>To search for a hair stylist in Lafayette, Louisiana, I will use the "Services_1_FindProvider" function with the required city parameter.</think> <tool_call>[{"name": "Services_1_FindProvider", "arguments": {"city" "Lafayette, LA"}}]</tool_call>

**Example 2: Multiple Function Calling**

Query: Imagine you are a drone operator. You are currently operating a drone that is at a point (5, 7) in the sky. You are asked to move the drone to a new point (10, 15). After reaching the new point, you are again asked to move the drone to another point (20, 25). Can you calculate the total distance the drone has traveled using the Euclidean norm method?

Output: <think>To calculate the total distance the drone has traveled, I will use the "math.hypot" function to calculate the Euclidean distance between each pair of points. First, I will calculate the distance from the initial point (5, 7) to the new point (10, 15). Then, I will calculate the distance from the new point (10, 15) to the final point (20, 25). The sum of these two distances will be the total distance traveled by the drone.</think> <tool_call>[{"name": "math.hypot", "arguments": {"x": 5, "y": 7}}, {"name": "math.hypot", "arguments": {"x": 10, "y": 15}}, {"name": "math.hypot", "arguments": {"x": 20, "y": 25}}]</tool_call>

