# OpenReview forum: "Nemotron-Research-Tool-N1: Exploring Tool-Using Language Models with Reinforced Reasoning"
_ICLR.cc/2026/Conference — ICLR 2026 Poster_

### Official Review · Reviewer_5Uca · 2025-10-30

**Soundness:** 3
**Presentation:** 3
**Contribution:** 3
**Rating:** 6
**Confidence:** 4

**Summary:**

This paper presents a family of open-source tool-calling LLMs trained via rule-based reinforcement learning (R1-style RL) rather than conventional supervised fine-tuning (SFT). The key innovation lies in using a binary reward that verifies only output format and tool-call correctness, omitting supervision on reasoning trajectories. Besides the main experiments, the authors further conduct systematic ablations, finding that pure RL outperforms SFT-then-RL, that binary rewards outperform fine-grained ones, and that structured reasoning tags (<think>, <tool_call>) are essential for generalization.

**Strengths:**

1. This paper demonstrates that pure rule-based RL can outperform traditional SFT and SFT+RL pipelines for tool-use, which is an important insight contradicting existing practice.

2. Tool-N1 models surpass GPT-4o and other closed- and open-source tool models (e.g., Hammer2.1, xLAM-2) on all tested benchmarks, validating the method’s effectiveness.

2. This paper includes ablations on reward granularity, data composition, model scaling, and backbone choice, offering valuable design guidance for future RL-based tool-use research.

**Weaknesses:**

1. The paper’s contribution is largely empirical; it lacks theoretical insight into why rule-based RL generalizes better than SFT or how binary rewards shape reasoning policies.

2. A binary {0,1} reward could cause credit assignment inefficiency or instability, yet no analysis or mitigation (e.g., shaping, curriculum) is discussed.

**Questions:**

No extra question. See weakness above.

---

> ### Author Response · Authors · 2025-11-20
> **Official Response to AnonReviewer 5Uca**
>
> We appreciate the insightful and detailed feedback! We also sincerely thank the reviewer for highlighting the strengths of our work, including the insight, performance and extensive ablations. We will address your questions in our following response.
>
> **[Re Weakness 1: Lacks theoretical insight into why rule-based RL generalizes better than SFT or how binary rewards shape reasoning policies.]**
>
>
> Thank you for the insightful comment. Following your suggestion, we provide theoretical reasoning below to clarify why rule-based RL with binary correctness rewards can yield stronger generalization than SFT. Before presenting the details, we briefly summarize the key distinction: during policy updates, SFT optimizes for imitating the reasoning trajectories in the training data, whereas rule-based RL optimizes directly for achieving task success. When the training and test distributions differ, RL tends to generalize better because it is driven by outcome-based optimization rather than trajectory imitation [1].
>
> We first denote the SFT objective. Let x be the input, y the demonstration trajectory, and \pi_\theta  the model policy. SFT maximizes the likelihood of reproducing demonstrated trajectories:
>
> **\theta_{\text{SFT}} = \arg\max_\theta \mathbb{E}_{(x,y)\sim D}[\log \pi_\theta(y|x)].**
>
> This objective encourages the policy to imitate demonstration actions. As a result, the learned policy tends to match the demonstrator’s behavior:
>
> **\pi_\theta(a_t|s_t) \approx \pi_{\text{demo}}(a_t|s_t).**
>
> We contrast this with RL. Let \tau denote a trajectory and R(\tau) a binary correctness reward. The RL objective maximizes the expected reward over sampled trajectories:
>
> **\theta_{\text{RL}} = \arg\max_\theta \mathbb{E}_{\tau\sim \pi_\theta}[R(\tau)].**
>
> This objective removes reliance on demonstration style and rewards only functional correctness. The corresponding policy-gradient update is:
>
> **\nabla_\theta J(\theta)
> = \mathbb{E}_{\tau\sim\pi_\theta}[R(\tau)\,\nabla_\theta \log \pi_\theta(\tau)].**
>
> Because only successful trajectories produce gradient signals, the policy is pushed toward actions that reliably increase success probability. This explanation aligns with the empirical generalization improvements observed in our results and other recent empirical study [1].
>
> *[1] Chu, T., Zhai, Y., Yang, J., Tong, S., Xie, S., Schuurmans, D., ... & Ma, Y. (2025). Sft memorizes, rl generalizes: A comparative study of foundation model post-training. ICML 2025.*
>
>
> **[Re Weakness 2: A binary {0,1} reward could cause credit assignment inefficiency or instability.]**
>
> Thanks for your insightful suggestions! We respectfully clarify that, in the context of RL training for tool-using LLMs, a binary correctness reward is in fact necessary for achieving strong performance. In this setting, credit should be assigned exclusively based on the final outcome of a tool call, namely, whether the produced call is fully executable and correct.
> As shown in Section 4.4.1 (Table 3), shaped or partial-credit reward variants consistently degrade performance. We include the extracted results below for convenience:
>
> | Reward Design                                          | BFCL-Live | ToolQA | BFCL-Non-Live |
> |--------------------------------------------------------|-----------|--------|-------------|
> | Binary correctness (ours)                              | **80.38** | **89.42** | **83.36**   |
> | Partial credit: correct tags                           | 76.61     | 87.70  | 81.52       |
> | Partial credit: correct tags + function name           | 77.08     | 88.14  | 81.90       |
> | Partial credit: tags + function name + argument keys   | 78.94     | 88.50  | 82.41       |
>
> These results demonstrate that reward shaping may push the model to optimize for intermediate patterns such as partial argument structures instead of focusing on the actual executable outcome.  In addition, the learning curves in Section 4.4 further show that Tool-N1 exhibits stable and reliable training behavior. We sincerely appreciate your suggestion regarding this!

---

> > ### Comment · Reviewer_5Uca · 2025-11-25
> > **Response**
> >
> > Thank you for yur response.
> >
> > The rebuttal clarified most of my confusion. I have raised the contribution score and I choose to maintain my original assessment of the overall score.

---

> > > ### Author Response · Authors · 2025-11-25
> > > **Response Appreciation**
> > >
> > > Thank you for the follow-up and for your careful evaluation! We sincerely appreciate your time and consideration.

---

### Official Review · Reviewer_Emqt · 2025-10-31

**Soundness:** 3
**Presentation:** 3
**Contribution:** 3
**Rating:** 6
**Confidence:** 2

**Summary:**

This paper presents Nemotron-Research-Tool-N1 (Tool-N1), a series of tool-calling language models trained using a rule-based reinforcement learning (RL) paradigm. The core contribution is the application of a "R1-style" RL approach, which uses a simple binary reward to judge the format and functional correctness of tool calls, without supervising the intermediate reasoning steps. The authors conduct a systematic empirical study, demonstrating that their method outperforms strong baselines, and intriguingly, that a pure RL training pipeline can surpass the commonly adopted "SFT-then-RL" paradigm in this domain.

**Strengths:**

- The paper effectively argues that supervised fine-tuning (SFT) on distilled reasoning traces can lead to "imitative" or "pseudo" reasoning. The proposed rule-based RL method is a compelling alternative that incentivizes the model to develop its own internal reasoning strategies, promoting better generalization. The motivation is clear and grounded in current limitations of the field.
- Strong Empirical Evidence.
- Rigorous and Insightful Ablation Studies.

**Weaknesses:**

- There is no computational effiency analysis. How fast is it running compared to baselines? Especially, how the speed becomes when the number of candidate tools becomes large? (since LLM receives the textual input with all the tools included)
- There should be some brief introductions to recent tool-calling baselines, either in Section 4.1 or a Related Work section.
- There is no public code or models which makes the experiment replication difficult.
- There is no explicit paragraph of major contributions in Introduction.

**Questions:**

- How the name "NEMOTRON-RESEARCH-TOOL-N1" is determined? There seems to be no evident hint in the paper. Considering to use a shorter name instead?
- How is the Reward Model trained? There should be some explicit introduction (even simply refering to some popular methods such as Bladley-Terry) to make the methodology self-contained.
- The training is conducted by GRPO. How would the performance change if using PPO instead? Is there any insights compared these two RL methods?

---

> ### Author Response · Authors · 2025-11-20
> **Official Response to AnonReviewer Emqt - Part 1**
>
> We sincerely appreciate your valuable time and insightful feedback. Thank you for highlighting the strengths of our work and providing so many detailed comments on our submission. We address each of your questions comprehensively in our responses below.
>
> **[Re Weakness 1: Need computational efficiency analysis.]**
>
> Thanks for the valuable suggestions. Computational efficiency is primarily determined by the (1) model architecture and (2) the length of the reasoning trace. Guided by the binary reward signal, Tool-N1 is optimized to produce only the minimal reasoning necessary for achieving the correct final tool call. As a result, it is expected to be more efficient than both its base model without RL training and other advanced models that do not constrain their reasoning chains.
> To validate this, we evaluate the same mocked tool-calling tasks using identical prompts across the strongest baseline (GPT-4o) in Table 1 and the corresponding base models with and without enforced reasoning before tool invocation. Under the same hardware condition, we measure the average inference time over ten runs for each model and show the results below.
>
> |            | 10 Tools | 30 Tools | 50 Tools |
> |------------|----------|----------|----------|
> | GPT-4o  (Non-Reason)    |  824 ms        |    902 ms      |     1014 ms     |
> | GPT-4o      |   858 ms       |     952 ms     |    1092 ms      |
> | Qwen 2.5-7b-Instruct (Non-Reason)      |   146.1 ms         |  150.1 ms         |     161.2 ms     |
> | Qwen 2.5-7b-Instruct      | 168.4 ms        |    196 ms     |    210 ms   |
> | Tool-N1-7B |     157.4 ms     |  165.4 ms        |   174.3 ms       |
>
> We do observe that under the same architecture, Tool-N1-7B is obviously better than its base models Qwen 2.5-7b-Instruct when both are required to generate reasoning before tool calling. Its efficiency is even comparable to the version without enforced reasoning. While Tool-N1 also appears more efficient than GPT-4o, we refrain from drawing conclusions from this comparison because Tool-N1 and closed-source models cannot be evaluated under identical hardware conditions with GPT-4o, making the results not directly comparable.
>
> **[Re Weakness 2 & 4: Writing.]**
>
> Thank you for the valuable suggestions. We have made revisions to the paper based on your feedback. Specifically:
>
> - In the Related Work section, we have included more recent studies on tool-calling baselines, providing a detailed analysis of their advantages and disadvantages, along with their relation to our work. (Line 708-728)
>
> - In the Introduction section, we have added a paragraph that highlights the main contributions of our work, presented in bullet points for clarity. (Line 102-110)
>
> Thank you again for your helpful suggestions on improving our writing!
>
> **[Re Weakness 3: Public code and models.]**
>
> Thank you for the suggestion. We absolutely agree with the importance of releasing code for reproducibility. Per ICLR’s double-blind policy, we cannot put the already published artifacts (e.g., URLs, GitHub repositories) that could reveal our identities during the review period. We will put the public repository URL to the full implementation in the camera-ready version in accordance with the policy. Thank you for the helpful feedback.
>
> **[Re Question 1: Why not use a shorter name?]**
>
> Thanks for the constructive suggestion. We agree that a shorter and clearer name would improve readability. As noted in the footnote on the first page, we use the shorter name Tool-N1 throughout the paper and will highlight this more clearly in the revision. We sincerely appreciate your feedback regarding this.
>
> **[Re Question 2: How is the reward model trained?]**
>
> Thank you for the question. We would like to clarify that we do not train or use any additional reward model in our pipeline. Instead, our method relies on a rule-based binary correctness signal, where the final tool-call outcome is deterministically checked and assigned a binary reward. We have clarified this in Section 3.3 and provided additional ablations in Section 4.4.1 to demonstrate the effectiveness of this design. Thanks for the suggestions.

---

> > ### Author Response · Authors · 2025-11-20
> > **Official Response to AnonReviewer Emqt - Part 2**
> >
> > **[Re Question 3: How is GRPO Comparing with PPO?]**
> >
> > | Non live        | Simple | Multiple | Parallel | Parallel Multiple | Overall |
> > |-----------------|--------|----------|----------|-------------------|---------|
> > | Tool-N1-7b-GRPO | 77.00  | 95.00    | 94.50    | 90.50             | 89.25   |
> > | Tool-N1-7b-PPO  | 74.92  | 95.00    | 90.00    | 88.50             | 87.10   |
> > | GPT-4o          | 79.42  | 95.50    | 94.00    | 83.50             | 88.10   |
> > | ToolACE-8B (FC) | 76.67  | 93.50    | 90.50    | 89.50             | 87.54   |
> > | Live            | Simple | Multiple | Parallel | Parallel Multiple | Overall |
> > | Tool-N1-7b-GRPO | 82.17  | 80.44    | 62.50    | 70.83             | 80.38   |
> > | Tool-N1-7b-PPO  | 79.46  | 78.92    | 68.75    | 66.67             | 78.68   |
> > | GPT-4o          | 84.88  | 79.77    | 87.50    | 75.00             | 79.83   |
> > | ToolACE-8B (FC) | 73.26  | 76.73    | 81.25    | 70.83             | 78.59   |
> >
> > Thank you for the valuable feedback. Following your suggestion, we added PPO as a new baseline and present the results here. As shown, standard PPO performs noticeably worse than both GPT-4o and Tool-N1-GRPO, and also underperforms Tool-ACE in the Non-Live setting. These results show that GRPO is an essential component of our method. The key insight is that GRPO typically outperforms PPO because it removes the value function and thus avoids value-estimation error, leading to more effective upd

---

> > > ### Author Response · Authors · 2025-11-26
> > > **Looking Forward to Your Reply**
> > >
> > > Dear Reviewer Emqt,
> > >
> > > Thank you again for your valuable time and constructive review. We tried our best to address the mentioned concerns. Your suggestions and feedback are very important to us. Since the discussion deadline is approaching, we would greatly appreciate it if you could take a moment to review our responses and let us know whether they sufficiently address your questions or if any points remain unclear. We are more than happy to provide additional clarification.

---

### Official Review · Reviewer_K5pn · 2025-11-01

**Soundness:** 3
**Presentation:** 3
**Contribution:** 3
**Rating:** 4
**Confidence:** 3

**Summary:**

The authors propose a rule-based reinforcement learning method to enhance the tool calling abilities of LLM based on GRPO. Their method uses a binary reward function with light supervision, e.g. it only checks functional correctness and format validity. They claim that it improves the generalization abilities vs conducting SFT (no experiments are provided to back up such claim). The authors present results that their method improves performances by conducting evaluation using a variety of tool calling benchmarks: BFCL, ACEBench and API bank. The authors also present a thorough set of ablations to validate their design choices.

**Strengths:**

- The authors present a very thorough set of ablations to validate their design choices:
- SFT vs RL vs SFT+RL,
- Scaling laws that showcase that using a more powerful model is better and good generalization accross different LLMs as backbones
- Ablation on reasoning format vs not using it
- Ablation on binary vs more fine grained reward function
- Ablation on data decomposition

- The paper is well written and easy to follow

**Weaknesses:**

- Novelty and originality: It is hard to assess the overall contribution since no other baselines are presented of works that provide ways to interweave reasoning with tool calling.

**Questions:**

- What percentage of data gets filtered out when you remove the samples with invalid tool calls?
- You claim throughout the paper that it improves reasoning and doesn't lead to pseudo-reasoning. However, there is no experiment to back up this claim. How can you show case that this is indeed the case experimentally?
- The QWEN model used is outdated, it would be good to present some results with more recent models.

---

> ### Author Response · Authors · 2025-11-20
> **Official Response to AnonReviewer K5pn - Part 1**
>
> We sincerely appreciate your valuable time and insightful feedback! We also thank the reviewer for highlighting the strengths of our work (i.e., the extensive experiments and the clarity of the paper writing, etc.). We address all concerns and questions in our detailed responses below.
>
> **[Re Weakness 1: Lack of baselines which provides ways to interweave reasoning with tool calling.]**
>
> Thank you for the valuable feedback. We would like to clarify that the DeepSeek-R1 results in Table 1 were obtained using interwoven reasoning with tool calling during the evaluation process. Following your professional suggestion, we have added two additional baselines that also employ interwoven reasoning with tool calling: the open-source Qwen3-14b/8b and the closed-source GPT-5 model. We present the results for both our method and the baselines below.
>
> |    Reasoning + Tool Callings        | Non Live  | Non Live | Non Live | Non Live          | Live   | Live     | Live     | Live              | Avg   |
> |-------------|-----------|----------|----------|-------------------|--------|----------|----------|-------------------|-------|
> |             | Simple    | Multiple | Parallel | Parallel Multiple | Simple | Multiple | Parallel | Parallel Multiple | \     |
> | GPT-5       |     72.5      |     86.5     |     82     |     77.5              |  77.52      |   67.43       |     68.75     |        58.33           |  78.8     |
> | DeepSeek-R1 | 76.4      | 94.5     | 90.1     | 88.0              | 84.1   | 79.9     | 87.5     | 70.8              | 80.88 |
> | Qwen3-14b  |    75.17       |      94.5    |      94    |   90.5        |    83.3    |   76.5       |   93.75   |    54.17           |   83.1    |
> | Qwen3-8b  | 77.5          |  94.5        |    90.5      |   91                |    80.2    |    77.6      |   81.25       |         70.8          |   83.2    |
> | Tool-N1-7B  | 77.0      | 95.0     | 94.5     | 90.5              | 82.2   | 80.4     | 62.5     | 70.8              | **84.82** |
> | Tool-N1-14B | 80.6      | 96.0     | 93.5     | 92.0              | 84.1   | 81.0     | 81.3     | 66.7              | **85.97** |
>
> From the table, we observe that Tool-N1 consistently outperforms all baselines under the interwoven reasoning–with–tool-calling setting. This aligns with our intuition, as Tool-N1 is specifically trained for tool-based reasoning and benefits from reinforced reasoning signals.
>
>
> **[Re Question 1: What percentage of data gets filtered out when you remove the samples with invalid tool calls?]**
>
> Thank you for the question. The table below provides the detailed data statistics, including the percentage of samples removed for each training dataset.
>
> |                    | Xlam  | ToolAce |
> |--------------------|-------|---------|
> | Raw Data           | 60000 | 11300   |
> | After Process      | 6000  | 9653    |
> | Removed Percentage | 0%    | 14%     |
>
> To ensure that SFT data can be reliably converted into verifiable RL data, we mainly discard samples that fail JSON parsing or exhibit formatting inconsistencies. We have made this explicit in Section 3.1. Thank you for the suggestion.
>
> **[Re Question 2:  Need experiment to back up the claims of pseudo reasoning.]**
>
> Thanks for your feedback. We mainly use the experiments in Section 4.3.2 to support this claim. We kindly clarify that the pseudo-reasoning mentioned in the paper refers to cases where SFT-trained models distilled from expert trajectories exhibit high trajectory-matching accuracy but still fail to produce correct executable tool calls [1][2]. To replicate this phenomenon, we distill trajectories from DeepSeek-R1 and then perform SFT on the same base model as Tool-N1. We include the corresponding results below for your convenience.
>
> |          | SFT on Reasoning Traj. | RL with Verifiable Reward |
> |----------|------------------------|---------------------------|
> | Non-Live | 87.54                  | 88.23                     |
> | Live     | 77.87                  | 78.24                     |
> | Avg.     | 82.71                  | 83.24                     |
>
> From the results, we observe that the proposed training framework yields substantially better tool-calling performance than simply mimicking distilled reasoning trajectories. We appreciate your suggestion and have further clarified and emphasized this point in Section 4.3.2.
>
> *[1] Chen, H., Tu, H., Wang, F., Liu, H., Tang, X., Du, X., ... & Xie, C. (2025). Sft or rl? an early investigation into training r1-like reasoning large vision-language models. arXiv preprint arXiv:2504.11468.*
>
> *[2] Chu, T., Zhai, Y., Yang, J., Tong, S., Xie, S., Schuurmans, D., ... & Ma, Y. (2025). Sft memorizes, rl generalizes: A comparative study of foundation model post-training. arXiv preprint arXiv:2501.17161.*

---

> ### Author Response · Authors · 2025-11-20
> **Official Response to AnonReviewer K5pn - Part 2**
>
> **[Re Question 3: The QWEN model used is outdated.]**
>
> Thank you for the question. Following the reviewer’s suggestion, we have added another recently open-sourced model, Nemotron-Nano-8B, to our evaluation, and the updated results are shown in the table below.
>
> | Base model | LLaMa 3 - 8B | LLaMa 3.1-8B | Nemotron-Nano-8B | Qwen 2.5-7B |
> |------------|--------------|--------------|------------------|-------------|
> | Non-Live   | 86.5         | 88.1         | 82.5             | 89.1        |
> | Live       | 65.8         | 76.2         | 75.2             | 80.2        |
> | Average    | 76.2         | 82.2         | 78.9             | **84.7**        |
>
> In addition, we would like to clarify that one of the main goals of this paper is to propose and validate the effectiveness of our RL training method, which is model-agnostic and does not rely on any specific backbone. To demonstrate this, we evaluate our method across multiple model families (LLaMA, Qwen, Nemotron) that span different architectures and training recipes. For example, Nemotron-Nano-8B adopts a hybrid Mamba + Transformer architecture.
>
> From the results, we observe that Qwen achieves the strongest performance compared with other base models. We ultimately chose Qwen because (1) it consistently outperforms other mainstream models in our evaluations, and (2) many recent works adopt Qwen as the base model for tool-calling and agentic tasks. We appreciate the reviewer’s suggestion regarding this choice.

---

> ### Author Response · Authors · 2025-11-25
> **Looking Forward to Your Reply!**
>
> Dear Reviewer K5pn:
>
> We sincerely appreciate your efforts in reviewing this paper! We tried our best to address the mentioned concerns. Your suggestions and feedback are very important to us. The discussion deadline is approaching now. Are there unclear explanations and descriptions here?
>
> We are highly encouraged if your concerns have been addressed. On the contrary, if you need any more clarification, we can provide it as soon as possible before the discussion deadline.
>
> Thanks!
>
> Authors

---

### Meta-Review · Area_Chair_dAGG · 2026-01-06

**Summary:**

The following five core concerns can be summarized as follows:
(1) Insufficient Baselines and Outdated Backbone Models. Reviewers point out the lack of baselines for reasoning-intertwined tool calling, and the Qwen model is outdated. It is necessary to supplement recent open-source/closed-source baselines and verify the model’s generalization across different backbones.
(2) Lack of Theoretical Support. The paper is mainly empirical and fails to theoretically explain why rule-based RL outperforms SFT.
(3) Insufficient Analysis. It lacks computational efficiency analysis and fails to investigate the potential inefficiency or instability of the binary reward in RL.
(4) Vague Methodological Details and Writing Issues. It does not fully introduce research related to tool calling, there is no explicit statement of key contributions in the introduction, and the code/models are not publicly available, which affects reproducibility.
(5) Unverified Key Claims. The claim that "RL avoids pseudo-reasoning" lacks experimental support.

**Reviewer Concerns:**

Reviewer Concerns Issues Potentially Addressed by the Rebuttal
1. Baselines and Backbone Models: Added baselines such as GPT-5 and Qwen3-8B/14B, supplemented experiments with recent models like Nemotron-Nano-8B, and verified the model’s generalization across different backbone families including LLaMA, Qwen, and Nemotron.
2. Theoretical and Reward Issues: Supplemented theoretical analysis of the objective functions of RL and SFT, proved that binary rewards outperform fine-grained rewards through ablation studies, and demonstrated training stability via learning curves.
3. Computational Efficiency Analysis: Added computational efficiency experiments (comparing inference time under different numbers of tools).
4. Methodological Details and Writing: Expanded the related work section, added key contributions in the introduction, promised to release code in the final version, and provided data filtering statistics (14% of the ToolACE dataset was removed, and 0% of the xLAM dataset was removed).

Outstanding Reviewer Concerns
1. Insufficient Generalization Capability: Newly added experiments on the Nemotron model show that the method’s performance fails to outperform most baselines, especially in the non-live setting.
2. Insufficient Theoretical Depth: The theoretical analysis is superficial. Although experiments prove the advantage of binary rewards, it does not deeply explore why binary rewards can avoid credit assignment issues.
3. Verification of Key Claims: Provided comparative experimental results between RL and reasoning trajectory SFT, but the improvement of RL over SFT is limited, making it difficult to prove that RL can avoid pseudo-reasoning.

**Reviewer Scores:**

- Reviewer K5pn: The reviewer may slightly raise score, but the concerns about SFT-induced pseudo-reasoning remain outstanding.
- Reviewer Emqt: The reviewer may slightly raise score because the concerns have been mostly addressed.
- Reviewer 5Uca: The reviewer may not change score because the concern about the lack of theoretical analysis on binary rewards remains outstanding.

---

### Decision · Program_Chairs · 2026-01-26

Accept (Poster)